# Novel Dual Incretin Receptor Agonists in the Spectrum of Metabolic Diseases with a Focus on Tirzepatide: Real Game-Changers or Great Expectations? A Narrative Review

**DOI:** 10.3390/biomedicines11071875

**Published:** 2023-07-01

**Authors:** Alexandros Leonidas Liarakos, Chrysi Koliaki

**Affiliations:** 1Centre for Diabetes and Endocrinology, Royal Berkshire NHS Foundation Trust, Reading RG1 5AN, UK; alexandros.liarakos@gmail.com; 2First Propaedeutic Department of Internal Medicine and Diabetes Center, Laiko General Hospital, Medical Faculty, National Kapodistrian University of Athens, 11527 Athens, Greece

**Keywords:** tirzepatide, glucose-dependent insulinotropic polypeptide, glucagon-like peptide 1, dual incretin receptor agonists, twincretins, type 2 diabetes mellitus, obesity, non-alcoholic fatty liver disease

## Abstract

The prevalence of metabolic diseases including type 2 diabetes (T2D), obesity and non-alcoholic fatty liver disease (NAFLD) increases globally. This highlights an unmet need for identifying optimal therapies for the management of these conditions. Tirzepatide is a novel dual incretin receptor agonist (twincretin) that activates both glucagon-like peptide-1 (GLP-1) and glucose-dependent insulinotropic polypeptide (GIP) receptors. The aim of this narrative review was to examine the impact of novel twincretins, focusing on tirzepatide, on the management of a wide spectrum of metabolic diseases. Data from preclinical and clinical trials have shown that twincretins significantly reduce blood glucose levels in T2D, and tirzepatide is the first agent of this class that has been approved for the management of T2D. Additionally, the beneficial impact of tirzepatide on weight reduction has been corroborated in several studies, showing that this agent can achieve substantial and sustained weight loss in obese patients with or without T2D. Data also suggest that tirzepatide could be a promising drug for hepatic steatosis reduction in individuals with NAFLD. The remarkable effects of tirzepatide on glycaemic control, weight loss and liver-related outcomes have posed new research questions that are likely to lead to further advancements in the treatment of T2D, obesity and related metabolic disorders.

## 1. Introduction

Diabetes mellitus (DM) represents a major public health issue that is characterized as a global pandemic and poses a significant burden upon healthcare systems and societies. The International Diabetes Federation (IDF) estimates that 537 million individuals worldwide are diagnosed with DM, and the figure is expected to rise by 50% over the next 25 years [1]. The continuous rise in the number of people living with DM and especially type 2 diabetes (T2D) is tightly associated with an increased incidence of obesity on a global scale [2]. In 2015, almost 604 million adults in more than 70 countries were estimated to be obese and this prevalence has more than doubled since 1980, reaching epidemic levels [3]. T2D and obesity have also been linked with numerous additional metabolic comorbidities including non-alcoholic fatty liver disease (NAFLD), which is now considered to be the leading cause of chronic liver disease across the globe, resulting in significant hepatic and extra-hepatic morbidity and mortality [4,5,6,7].

Optimal metabolic control and weight reduction achieved through lifestyle interventions including diet modification and physical activity represent the cornerstone of managing individuals with T2D, obesity and NAFLD [2,7]. Insulin resistance has been recognized as one of the fundamental pathophysiological mechanisms linking these conditions [4], but there are still many other questions to be answered. For instance, the pathogenesis of liver inflammation and fibrosis in patients with NAFLD remains incompletely understood and hence, there is currently no disease-specific treatment [6]. This highlights an unmet need for identifying optimal therapeutic options for the management of conditions related to metabolic dysfunction and their associated complications.

Incretin hormones are gut peptides that are released after nutrient intake and stimulate insulin secretion. Glucose-dependent insulinotropic polypeptide (GIP) and glucagon-like peptide-1 (GLP-1) are the known incretin hormones that are responsible for the incretin effect (a two- to three-fold higher insulin secretory response to oral as compared to intravenous glucose administration), which becomes blunted or even absent in patients with T2D [8]. Endogenous GLP-1 is secreted by intestinal L-cells, stimulating insulin synthesis and secretion along with β-cell proliferation, while it inhibits β-cell apoptosis and glucagon release from α-cells in the pancreas [9]. Apart from the regulation of glucose levels, GLP-1 has multiple other actions including slowing stomach emptying and gastrointestinal motility, a reduction in hepatic glucose production and steatosis as well as an increase in lipolysis in the adipose tissue and glucose uptake and glycogen synthesis in the muscles. In the cardiovascular system, GLP-1 improves myocardial contractility and endothelial function and reduces arterial stiffness, inflammation and blood pressure (BP). Other activities involve appetite suppression, increased satiety via action to the hypothalamus and neuroprotection in the brain as well as natriuresis in the kidneys [10]. Agonists of GLP-1 receptors (GLP-1 RAs) such as liraglutide, dulaglutide and semaglutide have been shown to achieve substantial glycaemic control and facilitate weight reduction, and have been used for the management of T2D and obesity with excellent results [11,12]. Endogenous GIP is secreted by intestinal K-cells, resulting in insulin and glucagon secretion, increased β-cell proliferation and reduced hepatic glucose production. Furthermore, it reduces gastric acid secretion and enhances triglyceride storage in adipose tissue by improving adipose tissue insulin sensitivity [10].

Tirzepatide is a novel dual incretin receptor agonist that activates both GLP-1 and GIP receptors, and has demonstrated very promising results in lowering blood glucose levels and reducing body weight [13,14,15,16]. Based on these effects, once-weekly subcutaneous tirzepatide received regulatory approval by the United States Food and Drug Administration (FDA) in May 2022 and marketing authorization by the European Commission in September 2022 for the treatment of patients with T2D as an adjunct to diet and exercise [17,18,19]. It also attained Fast Track designation as an anti-obesity medication independent of concurrent T2D in October 2022 [20]. Although the positive impact of these incretin-based agents has been widely described, it is yet to be clarified whether their beneficial effects on metabolic control can be entirely explained by weight loss or are mediated by direct effects and other independent factors.

The aim of the present narrative review was to examine the impact of novel dual incretin receptor agonists, focusing on the unimolecular co-agonist tirzepatide, on the management of a wide spectrum of metabolic diseases including T2D, obesity and NAFLD.

## 2. Literature Search Methodology

For the preparation of this narrative review, we applied the search terms “glucose-dependent insulinotropic polypeptide and glucagon-like peptide-1 receptor agonists”, “dual incretin receptor agonists”, “tirzepatide”, “LY3298176” and “twincretins” to retrieve available literature data from PubMed, Medline and Google Scholar from inception until May 2023. We included papers written in the English language that involved a synthetic methodology, reviews, in vivo studies and clinical trials in humans.

## 3. Rationale and Overview of Incretin Receptor Agonists

The gastrointestinal or elsewise gut-derived peptides are secreted by the enteroendocrine cells of the gastrointestinal tract after food ingestion and are well-established significant modulators of energy balance, glucose metabolism and eating behaviour. Aberrant alterations in their circulating levels seem to play an important role in the pathophysiology of obesity and T2D [21,22]. The best-studied gastrointestinal peptides are ghrelin, peptide tyrosine–tyrosine (PYY), GIP, GLP-1, oxyntomodulin and glicentin [23].

Therapeutic approaches targeting the incretin hormone receptors have been broadly investigated for treating individuals with T2D, obesity and metabolic syndrome [24,25]. A representative example widely used in clinical practice is GLP-1 RAs. GLP-1 RAs exert pleiotropic beneficial effects including glucose-dependent insulin secretion, inhibition of glucagon release, a delay of gastric emptying, appetite suppression and reduction of energy intake as well as an improvement in myocardial contractility and endothelial function [9]. Several once-weekly or once-daily GLP-1 RAs in oral or subcutaneous injectable formulations have been clinically applied including exenatide, liraglutide, lixisenatide, dulaglutide, albiglutide and semaglutide. The potent impact of these drugs on glycaemic control and body weight reduction in patients with T2D has been well documented in the literature [12,26,27,28]. Besides this class of medications, other single incretin receptor agonists include PYY, amylin and fibroblast growth factor 21 (FGF21) analogues.

Due to the possible side effects or counteradaptations induced by these single agents, unimolecular peptide polyagonists, which mimic closely the native gut hormone characteristics, have been recently developed and attracted considerable scientific interest. The underlying concept is that the combined activation of two or three gut hormone receptors may act synergistically, providing additive effects on lowering blood glucose and body weight in comparison to GLP-1 analogues alone. As a result, several preclinical and clinical trials have been designed and conducted, and have clearly shown substantial benefits of dual or triple gut hormone receptor agonists in reducing body weight, fasting and postprandial glucose levels as well as improving insulin sensitivity and serum lipid levels [29,30]. Dual incretin receptor agonists involve oxyntomodulin (OXM), glicentin and synthetic GLP-1/GIP, GLP-1/glucagon or GLP-1/amylin co-agonists. GLP-1/GIP/glucagon or GLP-1/OXM/PYY triagonists are agents that have also shown encouraging results in preclinical and clinical studies [31,32,33]. OXM is a peptide hormone released from L-cells after nutrient ingestion and represents a natural agonist of both GLP-1 and glucagon receptors. Its actions involve increased insulin and adiponectin secretion, enhanced lipolysis and hepatic glucose production as well as an increased energy expenditure accompanied by a reduced food intake, ghrelin secretion and gastric emptying [34]. Although the combining actions of GLP-1 and glucagon could make oxyntomodulin an effective treatment in obesity, its short plasma half-life of around 12 min in humans has limited its use in clinical practice [35]. Glicentin, whose 69 amino-acid sequence incorporates the sequence of OXM, is a proglucagon-derived hormone secreted by L-cells, and its roles and functions in humans are not completely understood [36]. Nielsen et al. have recently demonstrated that postprandial changes of glicentin and OXM could potentially predict the weight loss response after bariatric surgery [37]. Out of all these agents, tirzepatide (LY3298176, Mounjaro), a novel GIP/GLP-1 co-agonist (twincretin) that stimulates the GIP receptor more potently than the GLP-1 receptor [38], has emerged as a very promising drug in targeting metabolic dysfunction and has attracted a lot of interest in the literature [15,20,39]. The advantage of this synthetic GIP/GLP-1 co-agonist compared to OXM and glicentin is that GIP potentiates the satiety signal of GLP-1 and alleviates GLP-1-induced nausea [15].

Tirzepatide is a synthetic linear peptide composed of 39 amino acids. Some residues derive from GLP-1, GIP and semaglutide, while a few residues are distinct [40]. It includes a C20 fatty di-acid moiety and has a half-life of about 5 days, allowing once-weekly subcutaneous administration [41]. The commercially available tirzepatide delivery pen is an easy-to-use, single autoinjector with a hidden needle, ensuring patient satisfaction and optimizing treatment compliance. Preclinical data suggest that tirzepatide acts via several mechanisms of action including increased pancreatic β-cell proliferation and survival, increased insulin synthesis and secretion as well as a decreased appetite and food intake. It has also been associated with enhanced cardioprotection, decreased bone reabsorption and an improvement in cognitive function [42]. One should keep in mind that there is no available evidence to date in humans that tirzepatide is associated with increased β-cell survival, or that tirzepatide leads to cardioprotection. The outcomes of SURPASS-CVOT are eagerly awaited to reveal the impact of this synthetic twincretin on cardiovascular outcomes. Nevertheless, tirzepatide has demonstrated significant glycaemic efficacy and notable weight loss in patients with and without T2D [13,14,43,44,45].

## 4. Metabolic Effects of Tirzepatide

### 4.1. Impact of Tirzepatide on T2D and Glycaemic Control

The effect of twincretins on glycaemia has been explored in several animal studies. Irwin et al. evaluated the impact of subchronic (14 days) intraperitoneal administration of N-AcGIP (Lys37Myr), exendin(1–39)amide or the combination of both peptides in adult obese mice (ob/ob). Obese mice were administered an intraperitoneal injection of glucose alone or combined with GIP, N-AcGIP, GLP-1 or exendin(1–39)amide. N-AcGIP alone or in combination with exendin(1–39)amide significantly reduced non-fasting plasma glucose levels and glycated haemoglobin HbA1c [46]. Similar results were provided in an experimental study of Gault et al., who assessed the impact of liraglutide and N-AcGIP, a simple combination of liraglutide plus N-AcGIP or Lira-AcGIP preparation (Lira-AcGIP) in diabetic obese mice. The authors demonstrated that the Lira-AcGIP combination resulted in a significant improvement in glycaemic control compared to the other comparator interventions [47]. In 2018, Pathak et al. compared the acute glycaemic effects of Nac(D Ala2)GIP/GLP-1-exe, a peptide that binds to both GIP and GLP-1 receptors, with (D Ala2)GIP or exendin-4 therapy alone. The researchers showed that the twincretin approach could lead to a significant reduction in blood glucose levels at 4 and 8 h after the injection and was associated with an improved insulin secretion [48].

The beneficial glycaemic effect of twincretins has been further confirmed in phase 1 and 2 clinical trials in humans. In 2017, Schmitt et al. conducted a randomized, double-blind, phase 1 trial using NNC0090-2746, also known as RG7697, which is a GIP/GLP-1 dual agonist developed by Novo Nordisk. Patients with T2D were administered a once-daily subcutaneous injection of NNC0090-2746 (0.25–2.5 mg) or the placebo for 14 days. The data of this clinical study showed that the synthetic twincretin significantly reduced HbA1c, fasting and postprandial serum glucose levels in patients with T2D [49]. Frias et al. extended the characterization of NNC0090-2746 and performed a 12-week, randomized, placebo-controlled, double-blind, phase 2a trial, in which individuals with T2D who were inadequately controlled with metformin received either 1.8 mg of NNC0090-2746 or the placebo once daily subcutaneously. The authors have shown that NNC0090-2746 significantly improved glycaemic control compared with the placebo, and was generally safe and well tolerated [50]. These findings were followed by a double-blind, randomized, phase 2b study examining a novel GIP/GLP-1 co-agonist (LY3298176 or “tirzepatide”) developed by Eli Lilly and Company, Indianapolis, IN. In this trial, patients with T2D received either once-weekly subcutaneous tirzepatide (1 mg, 5 mg, 10 mg or 15 mg), dulaglutide (1.5 mg) or the placebo for 26 weeks. The analysis showed that tirzepatide significantly decreased HbA1c in a dose-dependent manner, while there were no reports of severe hypoglycaemia. However, in comparison to the 5 and 10 mg tirzepatide dosage groups as well as dulaglutide, the 15 mg group had a higher incidence of gastrointestinal adverse events including nausea, vomiting and diarrhoea and a higher rate of treatment discontinuation [51]. Based on these data, a 12-week, double-blind, placebo-controlled, phase 2 study was performed to further assess the efficacy and tolerability of higher doses of tirzepatide (12 and 15 mg) using three different dose-escalation regimens (12 mg (4 mg, weeks 0–3; 8 mg, weeks 4–7; 12 mg, weeks 8–11), 15 mg (2.5 mg, weeks 0–1; 5 mg, weeks 2–3; 10 mg, weeks 4–7; 15 mg, weeks 8–11) and 15 mg (2.5 mg, weeks 0–3; 7.5 mg, weeks 4–7; 15 mg, weeks 8–11)). The investigators demonstrated that tirzepatide resulted in clinically meaningful reductions in HbA1c, while lower starting doses and smaller dose increments were associated with a more favourable side effect profile [52].

In response to the promising outcomes derived from phase 1 and 2 clinical trials of twincretins, a series of multi-national pivotal phase 3 randomized controlled clinical trials (RCTs) in individuals with T2D evaluating the impact of tirzepatide on glycaemic control as the primary endpoint were conducted. These trials have been performed under the umbrella of the “SURPASS” programme and the major studies with published data are presented below and summarized in Table 1.

In SURPASS-1, patients with T2D inadequately controlled with diet and exercise alone were randomly assigned to receive a 40-week course of once-weekly tirzepatide (5, 10 or 15 mg) or the placebo. At 40 weeks, all tirzepatide doses were superior to the placebo for changes from baseline in HbA1c. Specifically, HbA1c decreased by 20 mmol/mol from baseline with tirzepatide, 5 mg; 21 mmol/mol with tirzepatide, 10 mg; and 23 mmol/mol with tirzepatide, 15 mg. Significantly more participants in the tirzepatide groups compared to the placebo group reached HbA1c glycaemic targets of less than 53 mmol/mol (87–92% vs. 20%) and 48 mmol/mol or less (81–86% vs. 10%), and 31–52% of patients on tirzepatide versus only 1% on the placebo reached an HbA1c level of less than 39 mmol/mol. Furthermore, there were no episodes of severe hypoglycaemia reported, while dose-dependent gastrointestinal events were the most frequent side effects [53]. In a post hoc analysis of SURPASS-1, tirzepatide monotherapy at doses 5, 10 and 15 mg was shown to induce significant improvements in several fasting biomarkers of pancreatic β-cell function and insulin sensitivity, effects which were only partially attributable to the observed weight loss, shedding more light into the underlying pathophysiological mechanisms explaining the improved glycaemic control of patients with T2D under tirzepatide treatment [54].

In SURPASS-2, participants with T2DM were randomized to receive tirzepatide (5, 10 or 15 mg) or semaglutide, 1 mg, once weekly for 40 weeks. The data showed that tirzepatide was non-inferior and even superior to semaglutide concerning the mean change in HbA1c from baseline to 40 weeks. The estimated differences between the 5 mg, 10 mg and 15 mg tirzepatide groups and the semaglutide group were −0.15 percentage points (*p* = 0.02), −0.39 percentage points (*p* < 0.001) and −0.45 percentage points (*p* < 0.001), respectively [55].

The SURPASS-3 trial evaluated the efficacy and safety of once-weekly tirzepatide versus once-daily titrated insulin degludec for 52 weeks in T2D individuals inadequately controlled with metformin, with or without sodium–glucose cotransporter 2 (SGLT2) inhibitors. From a mean baseline HbA1c of 8.17%, the estimated treatment difference versus degludec ranged from −0.59% to −1.04% for tirzepatide (*p* < 0.0001 for all tirzepatide doses), and the percentage of patients achieving HbA1c less than 7.0% (<53 mmol/mol) was greater (*p* < 0.0001) in all three tirzepatide dose groups (82–93%) compared to insulin degludec (61%) at week 52. In addition, fewer episodes of hypoglycaemia were reported in tirzepatide groups compared to degludec. The authors concluded that tirzepatide treatment can achieve better glycaemic control with a lower risk of hypoglycaemia in diabetic patients who are suboptimally controlled with oral glucose-lowering drugs compared to insulin degludec [56]. A substudy of the SURPASS-3 trial was designed (“SURPASS-3 CGM”), aiming to assess the efficacy of once-weekly tirzepatide versus once-daily degludec on glycaemic control measured by continuous glucose monitoring (CGM) in adults with T2D. The primary endpoint was the proportion of time spent in the tight target range (71–140 mg/dL) at 52 weeks in individuals treated with tirzepatide (10 and 15 mg) compared to degludec. The data showed that the participants receiving tirzepatide (pooled 10 and 15 mg groups) had a greater proportion of time in the target range compared with the subjects receiving degludec (estimated treatment difference, 25% (*p* < 0.0001)). These results provided further solid proof of the beneficial impact of tirzepatide on achieving glycaemic targets without an increase in hypoglycaemia risk when compared to basal insulin treatment [57].

In the SURPASS-4 trial, tirzepatide was compared with a different basal insulin, glargine, regarding the achievement of glycaemic control in people with T2D and increased cardiovascular risk being treated with metformin, a sulfonylurea or an SGLT-2 inhibitor. After 52 weeks of treatment, tirzepatide at all tested doses significantly reduced HbA1c compared with glargine. The investigators found that the estimated treatment difference versus glargine was −0.99% for tirzepatide, 10 mg, and −1.14% for 15 mg, and the non-inferiority criterion of 0.3% was met for both doses. Fewer hypoglycaemic episodes were also observed in the tirzepatide groups, and most importantly, all these outcomes were achieved with no excess cardiovascular risk [58].

SURPASS-5 was designed to assess the effect of once-weekly tirzepatide (5, 10 or 15 mg) compared with the placebo when added to titrated insulin glargine on glycaemic control in patients with T2D. After 40 weeks, the mean HbA1c change from baseline was −2.11% with 5 mg, −2.40% with 10 mg and −2.34% with 15 mg of tirzepatide vs. −0.86% with the placebo (*p* < 0.001 for all doses). These data further supported the evidence that tirzepatide is associated with a dose-dependent reduction in HbA1c in diabetic patients [59].

To understand the pathophysiological mechanisms underlying the action of tirzepatide in patients with T2D, Heise et al. conducted a randomized, double-blind, parallel-arm, phase 1 study in individuals with T2D treated with lifestyle measures and metformin, with or without one additional glucose-lowering drug. The participants were assigned to receive either tirzepatide (15 mg), semaglutide (1 mg) or the placebo once weekly for 28 weeks and the primary endpoint was the effect of tirzepatide vs. the placebo on the change in the clamp-derived disposition index (a composite outcome of insulin secretion and sensitivity). The authors found that the glycaemic benefits of tirzepatide in T2D resulted from concurrent improvements in key components of the diabetes pathophysiology, namely β-cell function, insulin sensitivity and glucagon secretion [60]. These effects could explain the remarkable glucose-lowering potential of tirzepatide observed in all phase 3 studies.

Lastly, the difference in T2D pathophysiology between Japanese and Caucasian individuals has been described previously in the literature [61]. To examine whether the glycaemic benefits of tirzepatide can also be implemented in Japanese patients with T2D, two phase 3 RCTs (SURPASS J-mono and SURPASS J-combo) were conducted. These studies have shown that tirzepatide is well tolerated and may significantly improve HbA1c when compared with dulaglutide, 0.75 mg, or the placebo, and demonstrated the applicability of tirzepatide to such patients regarding improvement in glycaemic control [62,63].

Taking all the above data into consideration, it becomes evident that tirzepatide displays a remarkable ability to lower blood glucose levels and even achieve normoglycaemia in patients with T2D, which allowed tirzepatide to be officially approved for the treatment of T2D in addition to diet and exercise in the USA and Europe [19].

### 4.2. Impact of Tirzepatide on Obesity and Weight Loss

Several animal studies have described the beneficial effects of twincretins on weight loss. A study of Irwin et al. showed that the combined administration of a GIP and GLP-1 receptor agonist in high-fat-fed (HFF) mice led to superior weight loss compared to GLP-1 RA alone [46]. This finding was further supported by Gault et al., who showed that both liraglutide and Lira-AcGIP could significantly reduce food intake in obese diabetic mice, but only Lira-AcGIP resulted in a significant decrease in body weight [47]. Other studies have also demonstrated that incretins combined together are able to reduce food intake and increase energy consumption in mice [46,64]. Pathak et al. assessed the metabolic effects of chronic treatment (twice daily for 28 days) with N-ac(D-Ala2) GIP/GLP-1-exe and exendin-4 alone, and in combination with (D-Ala2)GIP, in HFF mice. The investigators reported a significant decrease in body weight, which was mainly driven by a loss of the total body fat mass, while the lean body mass remained unchanged and was thus preserved [48].

Multiple clinical trials have robustly confirmed the powerful beneficial impact of twincretins in general and particularly tirzepatide on weight loss outcomes in humans [13,20,25,51]. A recent systematic review and meta-analysis, which included seven RCTs and a total of 6609 adults with T2D irrespective of their background glucose-lowering treatment, demonstrated a dose-dependent superiority of tirzepatide in reducing body weight compared to long-acting GLP-1 RAs, insulin or a placebo. Specifically, when compared to long-acting GLP-1 RAs (dulaglutide (1.5 mg) or semaglutide (1 mg) once weekly), tirzepatide resulted in larger reductions in body weight ranging from 1.68 kg with tirzepatide, 5 mg, to 7.16 kg with tirzepatide, 15 mg. Nevertheless, the authors acknowledged a series of study limitations, which mainly comprise the presence of statistical heterogeneity in the meta-analyses for change in HbA1c and body weight, the assessment of the risk of bias solely for the primary outcome (defined as change in HbA1c from baseline) and the limited generalization of the results mainly to individuals who are overweight/obese and already on metformin-based background treatment [13]. It should be also noted that weight loss was a secondary outcome in all these trials and results should therefore be interpreted with caution.

To date, there is only one RCT that investigated the effect of tirzepatide on weight loss as the primary outcome as summarized in Table 1 [43]. SURMOUNT-1 was a phase 3 double-blinded RCT, in which adults with a body mass index (BMI) of 30 or more, or more than 27 with at least one obesity-related complication excluding diabetes, were assigned to receive either once-weekly subcutaneous tirzepatide (at doses 5, 10 or 15 mg) or the placebo for a total duration of 72 weeks, including a 20-week dose-escalation period. The primary endpoints included the percentage change in body weight from baseline and a weight reduction of 5% or more, which is considered to be clinically significant. At baseline, the mean body weight was 104.8 kg, the mean BMI was 38.0 kg/m^2^ and 94.5% of participants had a BMI of 30 or higher. The data showed that, at week 72, the mean percentage change in body weight was −15.0% (95% CI, −15.9 to −14.2) with 5 mg of tirzepatide, −19.5% (95% CI, −20.4 to −18.5) with 10 mg and −20.9% (95% CI, −21.8 to −19.9) with 15 mg, whereas the respective weight change was only −3.1% (95% CI, −4.3 to −1.9) in the placebo group (*p* < 0.001 for all comparisons vs. placebo). The proportion of participants having a weight reduction of at least 5% was 85%, 89% and 91% with 5 mg, 10 mg and 15 mg of tirzepatide, respectively, and only 35% with the placebo. Interestingly, a substantial reduction in body weight of 20% or more was achieved in 50% and 57% of participants in the 10 and 15 mg of tirzepatide groups, respectively, compared with only 3% in the placebo group (*p* < 0.001 for all comparisons vs. placebo). As expected, the most common side effects were gastrointestinal adverse events mainly occurring during dose escalation [43].

Based on the clinical data described above, once-weekly subcutaneous tirzepatide can achieve substantial and sustained weight loss in obese patients with or without T2D and could thus represent a promising and powerful treatment for obesity in the near future.

### 4.3. Impact of Tirzepatide on NAFLD Outcomes

The association of NAFLD with clinically relevant extra-hepatic manifestations has been well described in the literature [6,65]. Younossi et al. have demonstrated a strong link between NAFLD and several cardiometabolic comorbidities including obesity, T2D, hyperlipidaemia, hypertension and metabolic syndrome as a constellation of risk factors [5]. These associations seem to be related either to the secondary effects of obesity or the direct pathophysiological effects of insulin resistance in NAFLD [6].

A recent systematic review and meta-analysis evaluating the global prevalence of NAFLD and non-alcoholic steatohepatitis (NASH) in the overweight and obese population has provided important data for improving understanding of the global NAFLD burden and optimizing disease management in this high-risk cohort of patients. In more detail, in the aforementioned analysis consisting of 101,028 individuals, the prevalence of NAFLD and NASH in the overweight population was found to be 70% and 33.5%, respectively. Similar prevalence estimates were reported in the obese population (75.3% for NAFLD and 33.7% for NASH, respectively). Clinically significant fibrosis (stages F2–4) was present in 20.3% of the overweight and 21.6% of the obese patients with NAFLD, while 6.7% of the overweight and 6.9% of the obese individuals with NAFLD presented with advanced stages of fibrosis (stages F3–4) [66]. Similarly, Ciardullo et al. showed that people with diabetes, and particularly T2D, not only have a higher prevalence of steatosis but also of significant liver fibrosis, as suggested by fibroscan results [67].

An additional meta-analysis has been performed to assess the global epidemiology of NAFLD/NASH in patients with T2D. According to this meta-analysis, the global prevalence of NAFLD and NASH in individuals with T2D was found to be 55.5% and 37.3%, respectively. The incidence of advanced fibrosis in patients with NAFLD and T2D was estimated to be 17.0% [68]. These figures indicate the clinical and economic burden of NASH in patients with T2D around the world and highlight the urgent need for optimal therapeutic options.

To address the need for NAFLD treatment in T2D patients, Hartman et al. performed post hoc analyses in a phase 2 trial exploring the effect of tirzepatide on biomarkers of NASH and liver fibrosis in T2D individuals receiving either once-weekly tirzepatide (1, 5, 10 or 15 mg), dulaglutide (1.5 mg) or the placebo for 26 weeks. Changes from baseline in aspartate aminotransferase (AST), alanine aminotransferase (ALT), procollagen III (Pro-C3), cytokeratin-18 (CK-18) and adiponectin levels were measured. The authors demonstrated a statistically significant decrease from baseline in AST (all groups except for tirzepatide, 10 mg), ALT (all groups), Pro-C3 (tirzepatide, 15 mg) and CK-18 (tirzepatide 5, 10, 15 mg) at 26 weeks. Tirzepatide at doses of 10 and 15 mg resulted in a significant reduction in CK-18, Pro-C3 and ALT levels when compared to the placebo and dulaglutide, respectively. In addition, tirzepatide at doses of 10 and 15 mg significantly increased adiponectin levels from baseline compared to the placebo. According to these data, higher doses of tirzepatide could significantly reduce NASH-related biomarkers and increase adiponectin in individuals with T2D [69]. Of note, there are currently no published data regarding the potential effects of tirzepatide on histological features assessed with liver biopsy such as hepatic inflammation and fibrosis in humans with NASH. To address this issue, there is an ongoing randomized, placebo-controlled, phase 2 clinical trial (SYNERGY-NASH; NCT04166773), which is expected to be completed within February 2024 and will likely provide strong evidence as to whether tirzepatide can reverse NASH and improve hepatic outcomes for patients with NAFLD.

Gastaldelli et al. have recently presented their findings from the SURPASS-3MRI trial, which was a substudy of the randomized, open-label, parallel-group, phase 3 SURPASS-3 trial [70]. To date, this has been the only study that investigated the impact of a 52-week treatment with once-weekly subcutaneous tirzepatide versus once-daily subcutaneous insulin degludec on liver fat content (LFC) in adults with inadequately controlled T2D and a fatty liver index of at least 60. As shown in Table 1, a total of 296 participants without a history of significant alcohol consumption were randomly assigned to receive active treatment (tirzepatide (5 mg), n = 71; tirzepatide (10 mg), n = 79; tirzepatide (15 mg), n = 72; insulin degludec, n = 74). The primary outcome was the change from baseline in LFC as evaluated with the MRI–proton density fat fraction (MRI-PDFF) at week 52, using pooled data from tirzepatide, 10 and 15 mg, vs. insulin degludec. The secondary outcomes included changes in the volume of visceral adipose tissue (VAT) and abdominal subcutaneous adipose tissue (ASAT). From an overall mean baseline LFC of 15.7%, at week 52, the absolute decrease in LFC was significantly greater for the pooled 10 and 15 mg of tirzepatide groups versus the insulin degludec group (−8.1% vs. −3.4%). The estimated treatment difference versus insulin degludec was −4.7% (*p* < 0.0001). Compared to baseline, after 52 weeks of treatment with tirzepatide, the mean LFC was reduced from 14.86 to 10.11 (tirzepatide, 5 mg), from 14.78 to 8.16 (tirzepatide, 10 mg) and from 16.65 to 8.59 (tirzepatide, 15 mg). The reduction in LFC was significantly correlated with baseline LFC, reductions in VAT, reductions in ASAT and reductions in body weight in the tirzepatide groups. The authors concluded that these results provide further evidence supporting the beneficial metabolic effects of this novel dual GIP/GLP-1 RA [70].

**Table 1 biomedicines-11-01875-t001:** Phase 3 randomized controlled clinical trials assessing the impact of tirzepatide, stratified by the primary outcome.

Study, Year of Publication	Population	Baseline Characteristics (Mean Values)	Sample Size and Study Groups	Primary Outcome	Results
Glycaemic control					
SURPASS-1, 2021 [53]	T2D patients inadequately controlled with diet and exercise alone and naive to injectable diabetes therapy	HbA1c, 7.9% (63 mmol/mol) Age, 54.1 years Women, 231 (48%) Diabetes duration, 4.7 years Weight, 85.9 kg BMI, 31.9 kg/m^2^	N = 478 Tirzepatide, 5 mg (n = 121) Tirzepatide, 10 mg (n = 121) Tirzepatide, 15 mg (n = 121) Placebo (n = 115) Duration: 40 weeks	Mean change in HbA1c from baseline at 40 weeks	HbA1c change: −1.87% (tirzepatide, 5 mg) −1.89% (tirzepatide, 10 mg) −2.07% (tirzepatide, 15 mg) +0.04% (placebo) (estimated treatment differences vs. placebo: −1.91% for tirzepatide, 5 mg; −1.93% for tirzepatide, 10 mg; −2.11% for tirzepatide, 15 mg (*p* < 0.0001 for all comparisons)) Weight change from baseline: −7.0 to −9.5 kg (tirzepatide groups) −0.7 kg (placebo)
SURPASS-2, 2021 [55]	T2D patients inadequately controlled with metformin, ≥1500 mg per day, for ≥3 months prior to screening	HbA1c, 8.28% (67 mmol/mol) Age, 56.6 years Women, 996 (53%) Diabetes duration, 8.6 years Weight, 93.7 kg BMI, 34.2 kg/m^2^	N = 1878 Tirzepatide, 5 mg (n = 470) Tirzepatide, 10 mg (n = 469) Tirzepatide, 15 mg (n = 470) Semaglutide, 1 mg (n = 469) Duration: 40 weeks	Mean change in HbA1c from baseline at 40 weeks	HbA1c change: −2.01% (tirzepatide, 5 mg) −2.24% (tirzepatide, 10 mg) −2.30% (tirzepatide, 15 mg) −1.86% (semaglutide, 1 mg) (estimated treatment differences vs. semaglutide: −0.15% for tirzepatide, 5 mg (*p* = 0.02); −0.39% for tirzepatide, 10 mg (*p* < 0.001); −0.45% for tirzepatide, 15 mg (*p* < 0.001)) Weight change from baseline: −7.6 to −11.2 kg (tirzepatide groups) −5.7 kg (semaglutide, 1 mg)
SURPASS-3, 2021 [56]	T2D patients inadequately controlled with metformin with or without SGLT2 inhibitors	HbA1c, 8.17% (66 mmol/mol) Age, 57.4 years Women, 635 (44%) Diabetes duration, 8.4 years Weight, 94.3 kg BMI, 33.5 kg/m^2^	N = 1437 Tirzepatide, 5 mg (n = 358) Tirzepatide, 10 mg (n = 360) Tirzepatide, 15 mg (n = 359) Degludec (titrated) (n = 360) Duration: 52 weeks	Non-inferiority of tirzepatide, 10 or 15 mg, or both vs. insulin degludec in mean change from baseline in HbA1c at week 52	HbA1c change: −1.93% (tirzepatide, 5 mg) −2.20% (tirzepatide, 10 mg) −2.37% (tirzepatide, 15 mg) −1.34% (degludec) (estimated treatment difference vs. degludec: –0.59% to –1.04% for tirzepatide (*p* < 0.0001 for all tirzepatide doses)) The non-inferiority margin of 0.3% was met for both types of doses Weight change from baseline: −7.5 to −12.9 kg (tirzepatide groups) +2.3 kg (degludec)
SURPASS-4, 2021 [58]	T2D patients with increased CV risk treated with metformin, a sulfonylurea or an SGLT2 inhibitor	HbA1c, 8.52% (70 mmol/mol) Age, 63.6 years Women, 749 (38%) Diabetes duration, 10.5 years Weight, 90.3 kg BMI, 32.6 kg/m^2^	N = 1995 Tirzepatide, 5 mg (n = 329) Tirzepatide, 10 mg (n = 328) Tirzepatide, 15 mg (n = 338) Glargine (titrated) (n = 1000) Duration: 52 weeks	Non-inferiority (0.3% non-inferiority boundary) of tirzepatide, 10 or 15 mg, or both vs. glargine in HbA1c change from baseline at 52 weeks	HbA1c change: −2.24% (tirzepatide, 5 mg) −2.43% (tirzepatide, 10 mg) −2.58% (tirzepatide, 15 mg) −1.44% (glargine) (estimated treatment difference vs. glargine: −0.80 for tirzepatide, 5 mg; −0.99% for tirzepatide, 10 mg; −1.14% for tirzepatide, 15 mg (*p* < 0.0001 for all comparisons)) The non-inferiority margin of 0.3% was met for both types of doses Weight change from baseline: −7.1 kg to −11.7 kg (tirzepatide groups) +1.9 kg (glargine)
SURPASS-5, 2022 [59]	T2D patients inadequately controlled with insulin glargine with or without metformin	HbA1c, 8.31% (67 mmol/mol) Age, 60.6 years Women, 211 (44%) Diabetes duration, 13.3 years Weight, 95 kg BMI, 33.4 kg/m^2^	N = 475 Tirzepatide, 5 mg (n = 116) Tirzepatide, 10 mg (n = 119) Tirzepatide, 15 mg (n = 120) Placebo (n = 120) Duration: 40 weeks	Mean change in HbA1c from baseline at week 40	HbA1c change: −2.11% (tirzepatide, 5 mg) −2.40% (tirzepatide, 10 mg) −2.34% (tirzepatide, 15 mg) −0.86% (placebo) (*p* < 0.001 for all comparisons with placebo) Weight change from baseline: −5.4 to −8.8 kg (tirzepatide groups) +1.6 kg (placebo)
SURPASS-3CGM, 2022 (Substudy of SURPASS-3) [57]	T2D patients, insulin-naive, treated with metformin alone or in combination with an SGLT2 inhibitor for ≥3 months before screening	HbA1c, 8.2% (66 mmol/mol) Age, 57.0 years Women, 110 (45%) Diabetes duration, 8.8 years Weight, 95.8 kg BMI, 33.9 kg/m^2^	N = 243 Tirzepatide, 5 mg (n = 64) Tirzepatide, 10 mg (n = 51) Tirzepatide, 15 mg (n = 73) Degludec (titrated) (n = 55) Duration: 52 weeks	Proportion of time that CGM values were in the tight target range (71–140 mg/dL) at 52 weeks (comparing pooled participants assigned to 10 and 15 mg of tirzepatide vs. degludec)	Patients on tirzepatide (pooled 10 and 15 mg groups) had a greater proportion of time in tight target range vs. degludec group (estimated treatment difference, 25% [95% CI, 16–33]; *p* < 0.0001)
SURPASS J-mono, 2022 [62]	Japanese T2D patients, treatment-naive or discontinued from oral antihyperglycaemic monotherapy	HbA1c, 8.2% (66 mmol/mol) Age, 56.6 years Women, 155 (24%) Diabetes duration, 4.8 years Weight, 78.2 kg BMI, 28.1 kg/m^2^	N = 636 Tirzepatide, 5 mg (n = 159) Tirzepatide, 10 mg (n = 158) Tirzepatide, 15 mg (n = 160) Dulaglutide, 0.75 mg (n = 159) Duration: 52 weeks	Mean change in HbA1c from baseline at week 52	HbA1c change: −2.40% (tirzepatide, 5 mg) −2.60% (tirzepatide, 10 mg) −2.80% (tirzepatide, 15 mg) −1.30% (dulaglutide, 0.75 mg) (estimated treatment differences vs. dulaglutide: −1.1% for tirzepatide, 5 mg; −1.3% for tirzepatide, 10 mg; −1.5% for tirzepatide, 15 mg (*p* < 0.0001 for all)) Weight change from baseline: −5.8 to −10.7 kg (tirzepatide groups) −0.5 kg (dulaglutide, 0.75 mg)
SURPASS J-combo, 2022 [63]	Japanese T2D patients inadequately controlled with oral antihyperglycaemic monotherapy for ≥3 months before screening	HbA1c, 8.6% (70 mmol/mol) Age, 57 years Women, 107 (24%) Diabetes duration, 8.6 years Weight, 77.5 kg BMI, 27.9 kg/m^2^	N = 443 Tirzepatide, 5 mg (n = 148) Tirzepatide, 10 mg (n = 147) Tirzepatide, 15 mg (n = 148) Duration: 52 weeks	Safety and tolerability during 52 weeks of treatment, assessed as the incidence of treatment-emergent adverse events in the modified intention-to-treat population	In total, 90% of participants completed the study and treatment In total, 77% of participants had ≥1 treatment-emergent adverse event HbA1c change from baseline: −2.50% (tirzepatide, 5 mg) −3.0% (tirzepatide, 10 mg) −3.0% (tirzepatide, 15 mg) Weight change from baseline: −3.8 to −10.2 kg
Body weight					
SURMOUNT-1, 2022 [43]	Adults with BMI ≥ 30 or BMI ≥ 27 and ≥1 weight-related complication excluding diabetes	Age, 44.9 years Women, 1714 (67.5%) Weight, 104.8 kg BMI, 38.0 kg/m^2^ Obesity duration, 14.4 years HbA1c, 5.6%	N = 2539 Tirzepatide, 5 mg (n = 630) Tirzepatide, 10 mg (n = 636) Tirzepatide, 15 mg (n = 630) Placebo (n = 643) Duration: 72 weeks	% change in weight from baseline at week 72 % of participants having ≥5% weight reduction from baseline at week 72	Weight change: −15.0% (tirzepatide, 5 mg) −19.5% (tirzepatide, 10 mg) −20.9% (tirzepatide, 15 mg) −3.1% (placebo) (*p* < 0.001 for all comparisons with placebo) Participants with ≥5% weight reduction: 85% (tirzepatide, 5 mg) 89% (tirzepatide, 10 mg) 91% (tirzepatide, 15 mg) 35% (placebo) (*p* < 0.001 for all comparisons with placebo) Participants with ≥20% weight reduction: 50% (tirzepatide, 10 mg) 57% (tirzepatide, 15 mg) 3% (placebo) (*p* < 0.001 for all comparisons with placebo)
Liver fat content					
SURPASS-3MRI, 2022 (Substudy of SURPASS-3) [70]	T2D patients with fatty liver index ≥ 60, (insulin-naive) treated with metformin alone or in combination with an SGLT2 inhibitor for ≥3 months before screening	LFC, 15.71% HbA1c, 8.2% (67 mmol/mol) Age, 56.2 years Women, 124 (42%) Diabetes duration, 8.3 years Weight, 94.4 kg BMI, 33.5 kg/m^2^	N = 296 Tirzepatide, 5 mg (n = 71) Tirzepatide, 10 mg (n = 79) Tirzepatide, 15 mg (n = 72) Degludec (titrated) (n = 74) Duration: 52 weeks	Change from baseline in LFC at week 52 (pooled data from the tirzepatide, 10 and 15 mg, groups vs. insulin degludec)	Change in LFC: −8.09% (pooled tirzepatide, 10 and 15 mg, groups) −3.38% (degludec) (estimated treatment difference vs. degludec −4.71% (95% CI, −6.72 to −2.70; *p* < 0.0001))

BMI: body mass index; CGM: continuous glucose monitoring; CI: confidence interval; CV: cardiovascular; HbA1c: haemoglobin A1c; LFC: liver fat content; SGLT2: sodium–glucose co-transporter-2; T2D: type 2 diabetes.

Considering all data presented above, tirzepatide appears to be a promising agent to reduce hepatic steatosis and may provide NAFLD patients with a novel therapeutic option as previously reported for selective GLP-1 Ras [71,72]. It is yet unknown how much of this benefit is attributable to weight loss per se, or whether there are any additional independent effects related to adipose tissue insulin sensitivity or whole-body systemic metabolism. Future research is required to address this question and elucidate the pathophysiological mechanisms underlying these effects, especially focusing on liver histology and a potential tirzepatide-induced improvement or reversal of histological features of NAFLD (inflammation, fibrosis).

## 5. Safety and Tolerability Issues

In general, the safety profile of tirzepatide was found to be comparable to that of the selective GLP-1 RAs, with the most common side effects being gastrointestinal complaints. In the placebo-controlled SURPASS trials, gastrointestinal side effects occurred in 37, 40 and 44% of patients treated with tirzepatide at 5, 10 and 15 mg, respectively [53,59]. The most common complaint was nausea, while other reported side effects included anorexia, vomiting, diarrhoea, constipation and abdominal pain [53,55,56,58,59,62,63]. Similar to the selective GLP-1 RAs, the majority of gastrointestinal events occurred in the period of dose escalation and were transient and of a mild and moderate severity. They tended to occur more frequently with the higher tirzepatide doses (dose-dependent effect). Discontinuation of treatment due to these side effects was more common with tirzepatide compared with a placebo [53,59]. In studies where tirzepatide was assessed as monotherapy or combined with metformin and SGLT2 inhibitors, the occurrence of clinically significant (level 2 or severe) hypoglycaemia was rare [53,55,56,62], while hypoglycaemia was more common when tirzepatide was added to a sulfonylurea or insulin [58,59,63]. With respect to vital signs, injection site and hypersensitivity reactions, diabetic retinopathy, thyroid malignancy and pancreatic and gallbladder outcomes, there were no alarming findings or clinically relevant safety signals across all phase 3 SURPASS trials [53,55,56,58,59,62,63]. Just like GLP-1 RAs, tirzepatide was shown to slightly increase heart rate, with increases in the mean pulse rate being dose-dependent and ranging from 1.1 to 2.9 beats per minute (bpm) with the 5 mg dose and 2.6 to 5.6 bpm with the 15 mg dose [53,55,56,58,59]. A consistent finding across all SURPASS trials was a dose-dependent reduction in systolic blood pressure (SBP) with tirzepatide, ranging from 4 to 12.6 mm Hg. In the two SURPASS trials comparing tirzepatide to a selective GLP-1 RA, the observed reduction in SBP was greater with tirzepatide than with dulaglutide (0.75 mg) and semaglutide (1 mg) [55,62]. Indeed, a recent meta-analysis across five SURPASS trials showed that tirzepatide can reduce SBP, an effect which was mainly related to weight loss. Specifically, the difference in mean SBP change from baseline at 40 weeks (total effect) between the tirzepatide and comparator groups was −1.3 to −5.1 mm Hg (tirzepatide, 5 mg), −1.7 to −6.5 mm Hg (tirzepatide, 10 mg) and −3.1 to −11.5 mm Hg (tirzepatide, 15 mg). Of note, SBP reduction was not dependent on antihypertensive medication use but rather on baseline SBP, which is reassuring regarding the theoretical risks of hypotension. The largest reduction in SBP was observed in the highest baseline category (>140 mm Hg), while those in the first quartile of the baseline SBP category (<122 mmHg) experienced no further decreases in SBP [73]. Injection site reactions were reported in 2.7% and hypersensitivity reactions in approximately 3.6% of tirzepatide-treated patients in the SURPASS trials. Nearly half of the patients developed anti-tirzepatide antibodies, which, however, had no effect on drug efficacy. Pancreatitis cases related to tirzepatide were rare, while cholelithiasis occurred in less than 1% of tirzepatide-treated patients despite substantial weight loss. The cardiovascular safety of tirzepatide was assessed in the SURPASS-4 trial as well as in a pre-specified meta-analysis of seven randomized controlled trials of greater than 26 weeks in duration [58,74]. In the latter meta-analysis, there was no increase in the risk of major adverse cardiovascular events with any tirzepatide dose compared to basal insulin glargine.

## 6. Knowledge Gaps and Areas for Future Research

It should be noted that at the time of writing this review, tirzepatide has not been compared in a head-to-head trial with higher doses of selective GLP-1 RAs such as dulaglutide, 4.5 mg, or semaglutide, 2 mg. There is also no head-to-head trial comparing tirzepatide with semaglutide, 2.4 mg, which is the dose that has been officially approved by the FDA for the indication of obesity. Recent data providing indirect comparisons suggest that tirzepatide at high doses (10 and 15 mg) may be more effective and tirzepatide at the dose of 5 mg may be comparable to higher-dose semaglutide in terms of HbA1c and body weight reduction, but there are currently no direct head-to-head trials comparing these agents [75,76]. Another critical question that remains unanswered is whether the impressive effects of tirzepatide on glycaemic control and body weight can actually translate into an improved cardiovascular prognosis by reducing cardiovascular mortality and other adverse cardiovascular outcomes in high-risk patients with T2D and obesity, providing solid evidence for clinically relevant cardioprotection. This intriguing question will be addressed by the SURPASS cardiovascular outcome trial (CVOT) (Lilly I8F-MC-GPGN; NCT04255433), which assesses tirzepatide versus dulaglutide in a large number of patients with T2D and established atherosclerotic cardiovascular disease. The completion of this trial and its preliminary results are anticipated within October 2024. Future clinical research should provide more evidence with regard to the safety and efficacy of tirzepatide in distinct patient populations including, for example, Black and Asian patients, patients with new-onset diabetes and children/adolescents, as well as in different clinical scenarios. Another critical issue is whether the unequivocal efficacy of tirzepatide in SURPASS trials can be extrapolated to efficacy in real-world clinical practice, considering that real-world clinical medicine can sometimes differ from the controlled settings of a pharmaceutical industry-sponsored clinical trial. In this regard, there should be an effort to minimize potential barriers to medication access and adherence in order to close the gap between clinical trials and real-world clinical practice efficacy. In the near future, real-world studies with tirzepatide will become available and provide valuable insights regarding the magnitude of tirzepatide effects in everyday clinical practice. Last but not least, an oral formulation of tirzepatide, maximizing patients’ compliance to treatment by overcoming the limitations of an injectable therapy, should also be tested in future clinical trials.

## 7. Conclusions

Tirzepatide is the first dual incretin (GIP/GLP-1) receptor agonist that has been approved for the management of patients with T2D. This once-weekly injectable agent has demonstrated remarkable results regarding glycaemic control and body weight reduction as well as encouraging outcomes for the treatment of NAFLD, as summarized in Figure 1. In addition to this potent impact, tirzepatide has revealed new aspects of the pathophysiology of diabetes and obesity, posing new research questions. For instance, elucidating how much of the benefit of tirzepatide is attributed to weight loss or any other independent effects will be very intriguing. One could also wonder whether adding a significant weight loss to excellent glycaemic control will change the progression and long-term prognosis of obesity, T2D and its associated micro- and macro-vascular complications. Several interesting research projects and studies will likely follow in the future to address these points. The next important step would be to identify the impact of tirzepatide on cardiovascular disease in patients with T2D for which the results of the SURPASS-CVOT are eagerly awaited. Lastly, tirzepatide seems to be only the beginning of dual incretin agonism since a number of different combinations, including unimolecular agonists for the GLP-1/glucagon, GLP-1/amylin and GLP-1/PYY receptors, are currently being studied. As a result, tirzepatide may be the first dual agonist to receive approval, but other drugs will likely follow and might potentially offer further advancements in the treatment of T2D, obesity and related disorders.

## Figures and Tables

**Figure 1 biomedicines-11-01875-f001:**
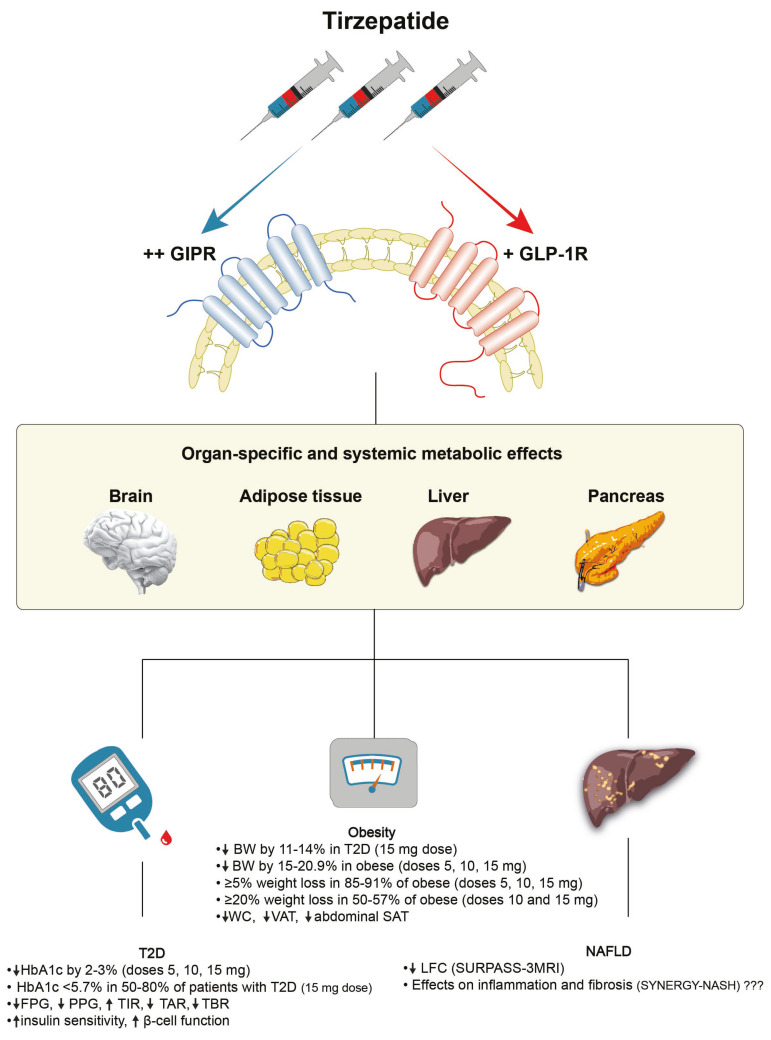
A summary of the beneficial effects of tirzepatide across the spectrum of metabolic diseases (T2D, obesity and NAFLD). BW: body weight; FPG: fasting plasma glucose; GIPR: glucose-dependent insulinotropic polypeptide receptor; GLP-1R: glucagon-like peptide-1 receptor; HbA1c: haemoglobin A1c; LFC: liver fat content; NAFLD: non-alcoholic fatty liver disease; PPG: postprandial glucose; SAT: subcutaneous adipose tissue; T2D: type 2 diabetes; TAR: time above range; TBR: time below range; TIR: time in range; VAT: visceral adipose tissue; WC: waist circumference.

## Data Availability

Not applicable.

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
