# Peer review of "Novel Dual Incretin Receptor Agonists in the Spectrum of Metabolic Diseases with a Focus on Tirzepatide: Real Game-Changers or Great Expectations? A Narrative Review"

_biomedicines, 2023, doi:10.3390/biomedicines11071875_

Round 1

Reviewer 1 Report

1.What is the advantage of using tirzepatide in comparison with native oxyntomodulin or any combinations of two incretin hormones?

2.What is known about protected forms of classic ambivalent incretins such as oxyntomodulin, glycentin etc.?

3.Authors reported about the clinical trial Eli Lilly based on comparative analysis of tirzepatide and dulaglutide. Which is the crucial advantage of tirzepatide in comparison with dulaglutide? Did HbA1c decreasing amplitude was significantly higher for tirzepatide in comparison with dulaglutide? In my opinion, only studies SURPASS-2 and SURPASS J-mono significantly confirmed that the tirzepatide can be a preferred option in glycemic control in comparison not with insulins but with real incretins analogues. 

4.The question is about NAFLD and tirzepatide therapy. Authors presented post hoc analysis data and announced the future histological results. Do authors know any MRT or other tomography data about the values hepatic density during tirzepatide or other incretin analogues therapies?

5.Do authors have any information about not only cardio, but on vascular diabetes complications, such as retinopathy, nephropathy, hindlimb ischemia, intermittent claudication etc in the tirzepatide treatment context? How tirzepatide can correct the grade of symptomatic clinical severity? I think for the clear incidence analysis the history of tirzepatide application is too small. In my opinion, it should be added in the manuscript.

Author Response

We thank all reviewers for their valuable suggestions, which helped us revise and improve our manuscript. We considered all their comments, performed all necessary revisions, and we provide below a detailed point-by-point response to every single comment, hoping to have adequately addressed all their remarks. All changes are denoted in red colour in the revised version of our manuscript.

Reviewer 1

1.What is the advantage of using tirzepatide in comparison with native oxyntomodulin or any combinations of two incretin hormones?

To address the valuable comment regarding the advantages of tirzepatide compared to natural coagonists, the sentence “The advantage of this synthetic GIP/GLP-1 coagonist compared to OXM and glicentin is that GIP potentiates the satiety signal of GLP-1 and alleviates GLP-1-induced nausea” (paragraph 3 of section 3, lines 141-143) has been introduced. Although the main focus of this review was to elaborate on the effects of tirzepatide on metabolic outcomes, we acknowledge that there are other promising combinations of incretin hormones such as GLP-1/glucagon as stated in detail in section 3. Additionally, in the conclusion section, we have acknowledged that tirzepatide seems to be only the beginning of dual incretin agonism since a number of different combinations, including unimolecular agonists for the GLP-1/glucagon, GLP-1/amylin, and GLP-1/PYY receptors, are currently being studied (lines 517-520).

2.What is known about protected forms of classic ambivalent incretins such as oxyntomodulin, glycentin etc.?

Thank you for your question. We have now included the following sentences in paragraph 3 of section 3 (lines 127-138): “OXM is a peptide hormone released from the L-cells after nutrient ingestion and represents a natural agonist of both GLP-1 and glucagon receptors. Its actions involve increased insulin and adiponectin secretion, enhanced lipolysis and hepatic glucose production as well as increased energy expenditure accompanied by reduced food intake, ghrelin secretion and gastric emptying. Although the combining actions of GLP-1 and glucagon could make oxyntomodulin an effective treatment in obesity, its short plasma half-life of around 12 minutes in humans has limited its use in clinical practice. Glicentin, whose 69 amino-acid sequence incorporates the sequence of OXM, is a proglucagon-derived hormone secreted by the L-cells, and its roles and functions in humans are not completely understood. Nielsen et al. have recently demonstrated that postprandial changes of glicentin and OXM could potentially predict weight loss response after bariatric surgery”.

3.Authors reported about the clinical trial Eli Lilly based on comparative analysis of tirzepatide and dulaglutide. Which is the crucial advantage of tirzepatide in comparison with dulaglutide? Did HbA1c decreasing amplitude was significantly higher for tirzepatide in comparison with dulaglutide? In my opinion, only studies SURPASS-2 and SURPASS J-mono significantly confirmed that the tirzepatide can be a preferred option in glycemic control in comparison not with insulins but with real incretins analogues.

As presented analytically in Table 1, the estimated treatment differences of tirzepatide vs dulaglutide in terms of HbA1c reduction were: -1.1% for tirzepatide 5 mg, -1.3% for tirzepatide 10 mg, and -1.5% for tirzepatide 15 mg (p<0.0001 for all). We absolutely agree that only the studies SURPASS-2 and SURPASS J-mono compared tirzepatide with other incretin analogues (semaglutide 1 mg in SURPASS-2 and dulaglutide 0.75 mg in SURPASS J-mono) and showed superiority of tirzepatide in terms of HbA1c reduction as depicted in Table 1. At the moment, there are no direct data from a head-to-head trial comparing tirzepatide with dulaglutide at higher doses (e.g. 1.5, 3 or 4.5 mg). The only study comparing tirzepatide with dulaglutide 1.5 mg is SURPASS-CVOT, whose main focus is cardiovascular endpoints, and its results are eagerly awaited within 2024.

4.The question is about NAFLD and tirzepatide therapy. Authors presented post hoc analysis data and announced the future histological results. Do authors know any MRT or other tomography data about the values hepatic density during tirzepatide or other incretin analogues therapies?

Thank you for your question/comment. We have now introduced the sentence “Compared to baseline, after 52 weeks of treatment with tirzepatide, the mean LFC was reduced from 14.86 to 10.11 (tirzepatide 5 mg), from 14.78 to 8.16 (tirzepatide 10 mg), and from 16.65 to 8.59 (tirzepatide 15 mg)” in paragraph 5 of section 4.3 to address your valuable comment (lines 412-414).

5.Do authors have any information about not only cardio, but on vascular diabetes complications, such as retinopathy, nephropathy, hindlimb ischemia, intermittent claudication etc in the tirzepatide treatment context? How tirzepatide can correct the grade of symptomatic clinical severity? I think for the clear incidence analysis the history of tirzepatide application is too small. In my opinion, it should be added in the manuscript.

Thank you for your comment. Currently, there is a paucity of clinical data on the impact of tirzepatide on diabetic retinopathy and peripheral arterial disease as well as improvement of their clinical severity. With regard to diabetic kidney disease, there is some evidence to suggest that tirzepatide could have reno-protective actions by reducing the risk of kidney-specific composite outcomes and worsening albuminuria (DOI 10.1016/j.metop.2023.100236, 10.1093/ckj/sfac274). We completely agree that the experience from the use of tirzepatide in clinical practice is currently limited and further research and real-world evaluation of its safety and efficacy is further required as acknowledged in section 6 (lines 488-499).

Reviewer 2 Report

I appreciate the work done by the authors since the role of novel dual incretin receptor agonists is of high importance in the cardiometabolic field. All sections are well written. In addition, table 1 and figure 1 are well prepared and very informative.

Minor comment

The present article lacks of full novelty, since many authors have already published similar critical review articles in the last months on the role of novel dual incretin receptor agonists on metabolic disorders, including diabetes, obesity and liver diseases. The authors need to be fully updated and also straight to the readers. Please briefly mention such previous publications, which largely overlap with the present article:

 - Glucagon-Like Peptide-1 Receptor Agonists and Dual Glucose-Dependent Insulinotropic Polypeptide/Glucagon-Like Peptide-1 Receptor Agonists in the Treatment of Obesity/Metabolic Syndrome, Prediabetes/Diabetes and Non-Alcoholic Fatty Liver Disease-Current Evidence. J Cardiovasc Pharmacol Ther. 2022;27:10742484221146371. doi: 10.1177/10742484221146371.

- The Emerging Role of Dual GLP-1 and GIP Receptor Agonists in Glycemic Management and Cardiovascular Risk Reduction. Diabetes Metab Syndr Obes. 2022;15:1023-1030. doi: 10.2147/DMSO.S351982.

- Tirzepatide: A novel, first-in-class, dual GIP/GLP-1 receptor agonist. J Diabetes Complications. 2022;36(12):108332. doi: 10.1016/j.jdiacomp.2022.108332.

Author Response

We thank all reviewers for their valuable suggestions, which helped us revise and improve our manuscript. We considered all their comments, performed all necessary revisions, and we provide below a detailed point-by-point response to every single comment, hoping to have adequately addressed all their remarks. All changes are denoted in red colour in the revised version of our manuscript.

Minor comment

The present article lacks of full novelty, since many authors have already published similar critical review articles in the last months on the role of novel dual incretin receptor agonists on metabolic disorders, including diabetes, obesity and liver diseases. The authors need to be fully updated and also straight to the readers. Please briefly mention such previous publications, which largely overlap with the present article:

We sincerely thank Reviewer 2 for the valuable comments. We acknowledge the high interest of clinicians and scientists in the impact of tirzepatide on metabolic outcomes, which has resulted in several relevant publications. We do believe that the importance of this topic justifies the publication of well-written and informative reviews providing readers with updated and concise information as well as emphasizing potential knowledge gaps and areas for future research. With this in mind, we produced this review, and we are grateful for the references suggested, which have been included in the revised version of our manuscript.

 - Glucagon-Like Peptide-1 Receptor Agonists and Dual Glucose-Dependent Insulinotropic Polypeptide/Glucagon-Like Peptide-1 Receptor Agonists in the Treatment of Obesity/Metabolic Syndrome, Prediabetes/Diabetes and Non-Alcoholic Fatty Liver Disease-Current Evidence. J Cardiovasc Pharmacol Ther. 2022;27:10742484221146371. doi: 10.1177/10742484221146371.

This reference was included in paragraph 3 of Introduction section.

- The Emerging Role of Dual GLP-1 and GIP Receptor Agonists in Glycemic Management and Cardiovascular Risk Reduction. Diabetes Metab Syndr Obes. 2022;15:1023-1030. doi: 10.2147/DMSO.S351982.

This reference was included in paragraph 4 of Introduction.

- Tirzepatide: A novel, first-in-class, dual GIP/GLP-1 receptor agonist. J Diabetes Complications. 2022;36(12):108332. doi: 10.1016/j.jdiacomp.2022.108332.

This reference was included in the last paragraph of section 3.

Reviewer 3 Report

In the present manuscript the Authors review the available preclinical and clinical evidence on the use of tirzepatide in the treatment of obesity, T2D and other metabolic conditions. The topic is certainly of interest and the manuscript is generally well-written and clear.

I have the following comments:

1-      I suggest to introduce a paragraph describing the activities of endogenous GLP1 and GIP at different organs in the introduction section before description of tirzepatide. One should also briefly describe what the incretin effect is.

2-      The authors state that “Tirzepatide acts via several mechanisms of action in- cluding increased pancreatic β cell proliferation and survival, increased insulin synthesis and secretion as well as decreased appetite and food intake. It has been also associated with enhanced cardioprotection, decreased bone reabsorption and improvement in cognitive function”. Please substantiate these statements and make it clear whether evidence was obtained in vitro, in animal models or in humans. For example there is no available evidence to date in humans that beta cell survival is increased, or that tirzepatide leads to cardio-protection or improvement in cognitive function. I suggest either removing this sentence or explaining that these results are still to be confirmed in humans (e.g. SURPASS-CVOT).

3-      What authors might emphasize more is the reduction in blood pressure that is seen with this agent, which seems to be related to body weight reduction.

4-      I suggest putting table 1 in a page in landscape orientation to improve readability, which is currently limited.

5-      When describing NAFLD prevalence and outcomes in patients with T2D or obesity, I suggest referring to a recent meta-analysis which clearly showed that patients with diabetes not only have higher prevalence of steatosis, but also of significant liver fibrosis, as suggested by fibroscan results (doi: 10.1016/j.diabres.2022.109981)

The manuscript is generally well-written, clear and easy to follow.

Author Response

We thank all reviewers for their valuable suggestions, which helped us revise and improve our manuscript. We considered all their comments, performed all necessary revisions, and we provide below a detailed point-by-point response to every single comment, hoping to have adequately addressed all their remarks. All changes are denoted in red colour in the revised version of our manuscript.

1-      I suggest to introduce a paragraph describing the activities of endogenous GLP1 and GIP at different organs in the introduction section before description of tirzepatide. One should also briefly describe what the incretin effect is.

Thank you for this productive comment. We have now introduced a whole new paragraph (paragraph 3 of Introduction section, lines 52-73) to address these points and provide the requested information.

2-      The authors state that “Tirzepatide acts via several mechanisms of action including increased pancreatic β cell proliferation and survival, increased insulin synthesis and secretion as well as decreased appetite and food intake. It has been also associated with enhanced cardioprotection, decreased bone reabsorption and improvement in cognitive function”. Please substantiate these statements and make it clear whether evidence was obtained in vitro, in animal models or in humans. For example there is no available evidence to date in humans that beta cell survival is increased, or that tirzepatide leads to cardio-protection or improvement in cognitive function. I suggest either removing this sentence or explaining that these results are still to be confirmed in humans (e.g. SURPASS-CVOT).

Thank you for this useful comment. We have now amended the last paragraph of section 3 to reflect your comment and we have included the following sentences for further clarification (lines 149-158): “Preclinical data suggest that tirzepatide acts via several mechanisms of action including increased pancreatic β-cell proliferation and survival, increased insulin synthesis and secretion as well as decreased appetite and food intake. It has been also associated with enhanced cardioprotection, decreased bone reabsorption and improvement in cognitive function. One should keep in mind that there is no available evidence to date in humans that tirzepatide is associated with increased β-cell survival, or that tirzepatide leads to cardio-protection. The outcomes of SURPASS-CVOT are eagerly awaited to reveal the impact of this synthetic twincretin on cardiovascular outcomes. Nevertheless, tirzepatide has demonstrated significant glycaemic efficacy and notable weight loss in patients with and without T2D”.

3-      What authors might emphasize more is the reduction in blood pressure that is seen with this agent, which seems to be related to body weight reduction.

Thank you for this suggestion. We appreciate the importance of addressing this point and as a result, the following sentences have been introduced in section 5 (lines 452-461): “Indeed, a recent meta-analysis across 5 SURPASS trials showed that tirzepatide can reduce SBP, an effect which was mainly related to weight loss. Specifically, the difference in mean SBP change from baseline at 40 weeks (total effect) between the tirzepatide and comparator groups was − 1.3 to − 5.1 mm Hg (tirzepatide 5 mg), − 1.7 to − 6.5 mm Hg (tirzepatide 10 mg) and − 3.1 to − 11.5 mm Hg (tirzepatide 15 mg). Of note, SBP reduction was not dependent on antihypertensive medication use but rather on baseline SBP, which is reassuring regarding the theoretical risks of hypotension. The largest reduction in SBP was observed in the highest baseline category (>140 mm Hg), while those in the first quartile of baseline SBP category (<122 mmHg) experienced no further decrease in SBP”.

4-      I suggest putting table 1 in a page in landscape orientation to improve readability, which is currently limited.

Thank you. We agree with your comment. Table 1 was put in landscape orientation to improve readability.

5-      When describing NAFLD prevalence and outcomes in patients with T2D or obesity, I suggest referring to a recent meta-analysis which clearly showed that patients with diabetes not only have higher prevalence of steatosis, but also of significant liver fibrosis, as suggested by fibroscan results (doi: 10.1016/j.diabres.2022.109981)

Thank you for providing this comment. We have now included the reference suggested and the sentence “Similarly, Ciardullo et al. showed that people with diabetes, and particularly T2D, not only have a higher prevalence of steatosis, but also of significant liver fibrosis, as suggested by fibroscan results” has been introduced in the second paragraph of section 4.3 (lines 367-369).

Reviewer 4 Report

Minor comment - Since several of the peptides mentioned in the manuscript are still considered "candidate" hormones, perhaps using the term "gastrointestinal peptides" is preferred over "gastrointestinal hormones"?  Again, very minor but physiologically relevant.

Otherwise, the work was well-written and comprehensive.  The discussion throughout was objective and the suggestions for future research are quite appropriate.

Author Response

We thank all reviewers for their valuable suggestions, which helped us revise and improve our manuscript. We considered all their comments, performed all necessary revisions, and we provide below a detailed point-by-point response to every single comment, hoping to have adequately addressed all their remarks. All changes are denoted in red colour in the revised version of our manuscript.

Minor comment - Since several of the peptides mentioned in the manuscript are still considered "candidate" hormones, perhaps using the term "gastrointestinal peptides" is preferred over "gastrointestinal hormones"?  Again, very minor but physiologically relevant.

Thank you for this comment. In response to your suggestion, we replaced the term "gastrointestinal hormones" by the term "gastrointestinal peptides" in the revised version of our manuscript (please see page 3, lines 97 and 101).

Round 2

Reviewer 1 Report

Many thanks to authors for the comprehensive response

Reviewer 3 Report

I find the manuscript significantly improved. I have no further comments.

The manuscript is generally well-written and easy to follow.